# Development of the Acoustic Comfort Assessment Scale (ACAS-12): Psychometric properties, validity evidence and back-translation between Spanish and English

**Karmele Herranz-Pascual** [1]⊙*, **Ioseba Iraurgi**[2]⊙, **Itziar Aspuru**[1]⊙, **Igone Garcia-Pérez**[1]⊙, **José Luis Eguiguren**[1]‡, **Álvaro Santander**[1]‡

**1** TECNALIA, Basque Research & Technology Alliance (BRTA), Derio, Spain, **2** DeustoPsych, Assessment, Clinical and Health, Faculty of Health Sciences, University of Deusto, Bilbao, Spain

⊙ These authors contributed equally to this work.
‡ JLE and ÁS also contributed equally to this work.
* karmele.herranz@tecnalia.com

**Data Availability Statement:** All relevant data are within the paper and its Supporting Information files.

## Abstract

A methodological proposal of a scale for human perception assessment of acoustic environment (acoustic comfort) in urban public spaces is presented: Acoustic Comfort Assessment Scale (*ACAS 12*). This paper shows the process of constructing this scale and its psychometric properties and validation. The approach is based on the soundscape conceptualisation collected in ISO 12913–1:2014 (Acoustics—Soundscape: Definition and conceptual framework). The proposed scale is a 5-point semantic differential scale made up of twelve pairs of bipolar adjectives, grouped around five theoretical dimensions. It is based on previous versions of 2-point and 3-point scales. The *ACAS-12* scale is the result of several empirical studies carried out by the authors on environmental and acoustic comfort assessment, beginning in 2011 in some cities of the Basque Country (Spain). Selected urban open places cover a wide variability of acoustic and non-acoustic characteristics, as well as the type of participants (real users) and activities carried out in these places. The original language of this scale is Spanish. The back-translation technique has been applied to create the English version. The results highlight the good psychometric properties of the *ACAS 12* scale (Cronbach's alpha 0.91 and composite reliability 0.90) and indicate that the best factorial solution is that of a single factor composed of the 12 pairs of adjectives that would explain approximately 50% of the *ACAS-12* variance (44% of extracted variance by Raykov's method or 56% by the MAP test). The absolute and incremental fit indices were above the minimum reference value of 0.90 and the residual-based indices showed values close to suitability (SRMR = 0.057; RMSEA = 0.042). This result supports the consideration of the *ACAS-12* scale as a general measure of acoustic comfort.

**Funding:** The author(s) received no specific funding for this work. However, we would like to point out that the data analysed in this paper correspond to those collected in several projects developed between 2011 and 2015 in which the scales analysed in this paper were applied. These projects are: 1) Sound Island Project, 2) QUADMAP Project (http://www.quadmap.eu/), 3) CITI-SENSE project (http://www.citi-sense.eu/), and 4) a project to assess the quality of the public squares in Sestao-Spain. These projects were performed with the financial support of the Bilbao City Council (1), the Basque Government (4), and the EU within the framework of the LIFE program LIFE 10/ENV/IT/407 (2), and the FP7 program (3).

**Competing interests:** The authors have declared that no competing interests exist.

## Introduction

There has been a prevailing trend of considering the urban acoustic environment only from the noise approach. As a consequence, some studies in this field have been mainly focused on its harmful effects on citizens. However, in recent years this tendency is changing and there are increasingly more studies that analyse the acoustic environment from a positive perspective as well, focusing their attention on the beneficial effects that it has on social and human behaviour [1–4]. This is the framework for the acoustic comfort approach, which is concerned with noise reduction, as well as the improvement and preservation of positive or pleasant acoustic environments [5–7]. For this reason, assessments that incorporate a measurement of the urban comfort of spaces for use and enjoyment are needed.

Within environmental comfort literature, there is no consensus on the definition of the concept of urban comfort and some of the suggested definitions are biased when considering comfort as the mere absence of discomfort [8], or because they are primarily focused on physical parameters [9–12]. However, other research has shown that reducing sound level does not necessarily lead to better acoustic comfort in urban areas [1, 7, 13, 14].

The authors contribute to this dialogue with the definition of a comfortable place as one that can generate a pleasant environmental experience for people and communities that carry out individual or social activities, especially those that involve social interaction [15]. To evaluate the acoustic dimension of urban environmental comfort, the use of the soundscape approach is considered suitable [16]. This approach has been developed within the framework of several European actions and projects, many of which formed part of the COST-Action TD 0804 on "Soundscape of European Cities and Landscapes". The soundscape approach analyses the sound environment from a holistic perspective that transcends the noise control approach. Its main characteristics are 1) transdisciplinarity; 2) multidimensional analysis; 3) multi-agent 4) sound as a resource; 5) emphasis on subjective data; and 5) evaluation based on different methods of assessment (triangulation) [5–7]. The soundscape approach is a flexible framework and studies that analyse the acoustic environment from a positive perspective, focusing on its quality and beneficial effects on people, have increased [17].

The key principles of the soundscape have been developed within ISO 12913 [18–20] to enable broad international consensus and provide a basis for communication across disciplines and professions that have an interest in soundscape or acoustic comfort [2, 5, 21]. Part 1 of the ISO 12913 [18] defines a conceptual framework. According to this standard, the soundscape is the acoustic environment as perceived or experienced and/or understood by a person or people in context. In other words, the soundscape is defined as the way people perceive, experience, or understand the acoustic environment in a physical environment [19]. Part 2 ISO 12913 [19] provides information about data collection in studies. Finally, Part 3 of ISO 12913 [20] provides requirements and supporting information on the analysis of data collected in-situ using the methods specified in Part 2 [19].

The goal of this paper is to present a proposal for measuring acoustic comfort (the Acoustic Comfort Assessment Scale—*ACAS-12*), which could contribute to the advancement of the evaluation and measurement of the human perception of the acoustic environment and its comfort. This scale has been developed in parallel with the progress and development of the aforementioned ISO soundscape standards. In this sense, the proposal included in this publication has similar but not identical attributes to the 'circumplex' pattern of the soundscape defined by the standard (ISO 12913-part 2 and 3). The main difference is that the 12 items of *ACAS-12* are structured around five dimensions. It applies the three dimensions already consolidated in the ISO (*Pleasantness*, *Eventfulness* and *Familiarity*), adding two others: *Informational capacity* and *Congruence*, considered by other research. Both scales are based on the

Semantic Differential, with minor differences in some of the adjectives used for the same attribute. This may be due to the different languages used (English or Spanish). The *ACAS-12* scale has been integrated into global studies that evaluate environmental comfort in urban places, providing within this framework the evaluation of acoustic comfort in these places.

The best place for the application of *ACAS-12* is urban settings given the current global trend towards urbanisation [22]. It is considered that the most suitable solution to evaluate urban open environments is to carry out field studies involving people who use those spaces. For this, the tool for collecting data from people is a questionnaire in which acoustic comfort is evaluated using a Semantic Differential scale. These field studies offer high ecological validity [19, 23–25].

The Semantic Differential method has been used for acoustic environmental studies since the 70s [26] and therefore there is a high variability of dimensions (and pairs of adjectives) proposed to evaluate the soundscape [1, 20, 27, 28].

Until now, the main challenge when assessing acoustic comfort has been that it is a multifaceted phenomenon and hence cannot be measured using a few simple numbers [2]. Consequently, in the proposed methodology the main issue is to determine the set of bipolar adjectives that best describes urban soundscape, overall or in its different dimensions. The main highlights of the literature consulted are shown below.

The most extensively analysed dimension and with greater consensus is *Pleasantness*. This is used in questionnaires as well as in soundwalk and narrative methods. Kang and Schulte-Fortkamp [2] refer to the soundscape as an integral blend of the dimensions of *displeasure-pleasure* (valence) and *passive-active* (arousal), thereby introducing a second dimension subsequently identified as *Eventfulness*. These two dimensions, *Pleasantness* and *Eventfulness* (or *calmness* and *vibrancy* for Cain et al. [1]), are conceived as orthogonal components, organising the soundscape attributes in a circular or 'circumplex' pattern [2, 20, 28] defined by the following attributes: *pleasant*, *vibrant*, *eventful*, *chaotic*, *annoying*, *monotonous*, *uneventful* and *calm* as eight vectors separated by 45˚ in a circumplex model of the soundscape. These eight attributes are included in the scale of Perceived Affective Quality of Sound set out in ISO 12913–2:2018 standard [19]. In this two-dimensional space, a *vibrant* soundscape would be both *pleasant* and *eventful*, whereas a *calm* soundscape would be pleasant and uneventful. Correspondingly, a *chaotic* soundscape would be *unpleasant* and *eventful*, whereas a *monotonous* soundscape would be *unpleasant* and *uneventful*. The generality of the two-dimensional model is still under examination [29] and it requires further validation across languages and places [20].

Some authors identify two other dimensions: *Familiarity* explained by *common*, *familiar*, *real* and *rare* [28, 30] and *Informational capacity* of the sound [26, 28]. Moreover, field studies in urban parks and open green spaces suggest that informational soundscape properties (i.e., sound categories) are better predictors of soundscape quality than acoustic environmental measurements, such as the equivalent sound pressure level, LAeq [26, 31].

Previously, Raimbault et al. [32] obtained three factors. The first coincides with *Pleasantness*, while the second could be compared to *Eventfulness*. Nevertheless, the third is associated with the *Spatial dimension* with descriptors such as *organised–disorganised* or *nearby–far*.

Moreover, there is another interesting aspect when defining the soundscape and acoustic comfort, which is its *Congruence* in the perception of the acoustic environment with the global perception of the place landscape [33–36].

To identify the soundscape dimensions, researchers carried out factor analysis [20]. The result of this analysis shows that there is a main factor related to *Pleasantness* that accounts for more than half of the explained variance (in the review by Axelsson et al., [28]). The factor related to *Eventfulness* accounts for 15–20% of the variance, and *Familiarity* and *Informational*

*capacity* for 5–10%. However, in some research [34] the different components of the soundscape are grouped around a single general dimension that gathers the different theoretical components of the Perceived Restorativeness Soundscape Scale (fascination, being-away, compatibility, and extent coherence and scope).

Therefore, this paper aims to present a proposal for a semantic differential scale to assess acoustic comfort. It is the *ACAS-12* scale that includes 12 pairs of adjectives and that collects in an integrated way five dimensions of the soundscape that have been highlighted in the literature. Those dimensions are *Pleasantness*, *Eventfulness* and *Familiarity*, as well as *Informational capacity* and *Congruence* between soundscape and landscape. The proposed scale has been designed for on-site application in field studies involving real users to assess urban open public spaces.

The goal of this research is to contribute to the understanding of how an urban soundscape can create a pleasant environmental experience for people. This experience is called acoustic and urban comfort and can contribute to the general well-being and health of the population. In this way, the relevant cultural functions of these social ecosystems could be improved.

## Methods

This section presents the proposal for the acoustic comfort assessment tool, *ACAS-12* as well as its development process. A stage-based approach is proposed for such development. In stage 1, a 2-point previous version of the scale -*SSC-2A*- was used (2011–2012). Stage 2 shows how the 3-point previous version of scale -*SSC-3A*- was applied (2012–2013). And finally, in stage 3 the version of 5-point scale -*ACAS-12*- was used (2014–2015). The different studies carried out in these three stages are presented. A brief description of the characteristics of the sample and the campaign procedure is included. The scale used in each stage is evaluated through its psychometric characteristics. In this sense, the analysis of the data set of each scale is presented, which allows its evaluation and the decisions taken to define the final proposed tool. Finally, the proposed assessment scale is presented, both in the native language version (Spanish) and in the English version.

### Assessment tool: Acoustic Comfort Assessment Scale (*ACAS-12*)

The acoustic comfort measurement tool proposed in this article is the Acoustic Comfort Assessment Scale (*ACAS-12*), which is a 5-point semantic differential scale structured around 12 pairs of bipolar adjectives. The pairs of adjectives used in the proposal are shown in Table 1 and cover the five dimensions selected from the literature: *Pleasantness*, *Eventfulness* and *Familiarity*, as well as *Informational capacity* and *Congruence*.

Adjectives and dimensions are defined using different terms. The dimension <u>*Pleasantness*</u> has been defined by the pair of adjectives *unpleasant-pleasant*. In addition, two other pairs have been added: *artificial-natural*, due to the influence of nature in the global pleasant experience in a place (based on the literature on the benefit provided by contact with nature); and *stressful-relaxing*, related to its effect on perceived health. <u>*Eventfulness*</u> is defined by the *monotonous-lively* pair of adjectives. The interaction between these dimensions would be determined by two pairs of adjectives: *boring-fun* and *noisy-calm*. As explained earlier in the introduction, this scale was developed at the same time as the soundscape standardization was carried out. In this sense, the soundscape 'circumplex' pattern that has been defined in the standard (ISO 12913-part 2 and 3) and the *ACAS-12* have similar but not identical attributes [19, 20]. The pairs of adjectives defined in the standard are *pleasant*, *fun*, *lively*, *noisy*, *unpleasant*, *boring*, *monotonous*, and *calm*.

**Table 1. List of the 12 bipolar adjectives of the Acoustic Comfort Assessment Scale (*ACAS-12*) in Spanish (native language) and English.**

| Dimensions | | Adjectives | | | |
|---|---|---|---|---|---|
| | | English | | Spanish | |
| Pleasantness | 1 | unpleasant | pleasant | desagradable | agradable |
| | 2 | stressful | relaxing | estresante | relajante |
| Eventfulness | 6 | artificial | natural | artificial | natural |
| | 7 | monotonous | lively | monótono | vibrante |
| Pleasantness & Eventfulness | 3 | noisy | peaceful, calm | ruidoso | tranquilo |
| | 5 | boring | fun | aburrido | divertido |
| Familiarity | 8 | unknown | familiar | desconocido | familiar |
| | 9 | intermittent | uninterrupted | discontinuo | continuo |
| Information | 10 | hinders conversation | facilitates conversation | dificulta conversación | facilita conversación |
| | 11 | no informative | informative | no informativo | informativo |
| Congruence | 4 | chaotic, confusing | clear, accurate | caótico, confuso | claro, nítido |
| | 12 | inappropriate for the surroundings | appropriate for the surroundings | inapropiado con su entorno | apropiado con su entorno |

Moreover, ACAS-12 includes the dimension of *Familiarity*, which is described with two other pairs of adjectives: *unknown-familiar*; and *intermittent-uninterrupted*. And it also includes the dimension of *Informational-capacity* that is characterised by three pairs of adjectives: *hinders conversation-facilitates conversation*; *not informative-informative*; and *chaotic-clear*. Finally, the dimension of *Congruence* is defined with a single pair of adjectives: *inappropriate for the surroundings-appropriate for the surroundings*.

*ACAS-12* seeks a clear understanding of the concepts and therefore considers two optional adjectives to define each opposite term of the *Informational-capacity* dimension: *chaotic or confused* versus *clear or accurate*. In the Spanish version, *ACAS-12* also proposes two optional adjectives for the concept of *vibrancy*, where the positive adjective is defined as '*vibrante*' or '*animado*'.

The tool has been initially designed in Spanish because it is the native language of the participants in the studies carried out. An English version of the scale has been developed using the back-translation technique described below. Consequently, although there are concepts in common with other researchers, some attributes are represented by different pairs of adjectives. This is because in the back-translation process the most suitable Spanish adjective derived from English adjectives different from those considered by other researchers.

**ACAS–12 definition process.** The *ACAS-12* was designed in a three-stage process. The tool was improved considering the analysis of the data obtained in each stage which is shown later.

In stage 1 (2011–2012) the scale used, called *SSC-2A*, is a list of 22 unpaired positive and negative unmatched soundscape attributes (11 pairs). It was not a scale as such, since the participants were asked to choose those attributes that, from their point of view, best described the acoustic environment they perceived in the places they were using. They were also asked to differentiate how appropriate the perceived sound (12 pairs) was to the environment where it occurred (congruence) on a 5-point ordinal scale (1 very inappropriate, 2 inappropriate, 3 neither appropriate nor inappropriate, 4 appropriate and 5 very appropriate).

In stage 2 (2012–2013) the scale used, *SSC-3A*, was an improvement on the previous one. The same 11 pairs of adjectives were assessed but on a semantic differential scale of 3-point. The congruence was still independently rated using a 5-point scale.

In stage 3 (2014–2015), the proposed *ACAS-12* was used. As explained before, it is a semantic differential scale of 5 points. The pairs of adjectives were reviewed, with some small changes

in their names and the congruence of the perceived acoustic environment with the environment where it occurred was included as another pair of bipolar adjectives (Table 1).

## Case studies

The case studies used in this process were five environmental and acoustic comfort campaigns carried out from 2011 to 2015, in which 23 urban places from 10 public spaces located in Bilbao, Vitoria-Gasteiz and Sestao (Basque Country, Spain) were analysed.

1. Paseo_del_Arenal (stage 1: scale *SSC-2A*): It is a pedestrian urban square located in Bilbao's historical district. This space was subdivided into three areas (place 1, place 2, and place 3) for the analysis, differentiated by their functionality and their acoustic characteristics.

2. Plaza_Nueva (stage 1: scale *SSC-2A*): It is a pedestrian urban square located in Bilbao's historical quarter surrounded by arcaded buildings. This space was subdivided into three areas for the analysis, differentiated by their functionality and use: place 4 -covered arcades-; place 5 -terraces- and place 6 -square centres-.

3. Plaza_Levante (stage 2: scale *SSC-3A*): It is a small square close to an underground entrance in a neighbourhood of Bilbao (place 7).

4. Plaza_San_Pedro_Deusto (stage 2: scale *SSC-3A*): The place is located near the church in another neighbourhood of Bilbao (place 8).

5. Plaza_General_Latorre (GLT): It is an urban square located in another neighbourhood of Bilbao. It was renovated to make it the city's first sound island. The place was evaluated on two occasions: before renovation applying scale *SSC-2A* and being placed at 9 GLT_pre (stage 1). After the renovation of the square, the *ACAS-12* scale was used, being placed at 10 GLT_post (stage 3).

6. Calle_Los_Herrán (state 3: scale *ACAS-12*): It is the area where the central bus station of the city of Vitoria was previously located. This area includes place 11 -near the bus-stop-; place 12; and place 13.

7. Parque_Salinillas_de_Buradón (state 3: scale *ACAS-12*): This park is located in a new urban area of Vitoria and is on a small hill, close to the city's green belt. This area includes places 14 and 15.

8. Plaza_Constitución (state 3: scale *ACAS-12*): This public space is next to the northern entrance to the city of Vitoria. This area includes places 16 and 17.

9. Parque_Olarizu (state 3: scale *ACAS-12*): This park is part of the green belt of Vitoria and the Environmental Research Centre (CEA) is located there, receiving thousands of visitors during the year. This area includes place 18 –path-, and place 19.

10. Plaza-San-Pedro-Sestao (state 3: scale *ACAS-12*): This place is one of the most used urban public spaces in Sestao. This area includes place 20, place 21, place 22 -south zone with benches-, and place 23 –the rest zone-.

The places analysed encompass a variety of open urban public spaces. They have diverse acoustic environments, users, activities, times of the year and times of the day. From an acoustic point of view, the places ranged from very noisy (Plaza_Nueva-covered arcades, Sestao-rest zone with an equivalent sound level LAeq above 69 decibels dBA) to very quiet (Parque_Salinillas, with LAeq of 51 dBA). The presence of water, green areas or architectural elements are identified in each place. Regarding their use, the selected places are used for walking,

socializing, relaxing, passing through, etc., and for children to play. The participant samples are also very different socio-demographically, both generationally and concerning educational level or employment status. Furthermore, all samples are gender balanced.

## Campaign procedure

In general, the procedure followed for data collection in the different studies was similar. Each assessment of each space was carried out in the same place and analysed by a person who was using the space and who was invited to participate in the study. Data collection was carried out at times of the day when the public used the space most frequently. The participants were interviewed by qualified personnel previously trained in conducting social surveys. Furthermore, those responsible for the campaigns subsequently carried out an exhaustive inspection of each questionnaire completed by the interviewers and of each of the data collection campaigns. Therefore, there is a high level of certainty about the quality of the data collected in the analysed database.

The data presented in this study were collected through surveys of the users of the settings evaluated, following the ethical guidelines applicable at the time in our country Spain and Europe. At the time of the survey campaigns (2011–2015), the study didn't need to be approved by an institutional review board or ethics committee, so the current requirements were not applicable.

The participants agreed to take part in the projects, responding to the questionnaire, after having been informed following the procedures established by the laws in force in Spain and the Basque Country on the protection of personal data and statistical confidentiality.

Each database with the acoustic comfort assessments carried out in each of the 23 urban places was cleaned-up. The total sample is 593 valid surveys, of which 263 (44.3%) were carried out with the first version of the scale, *SSC-2A*; 84 surveys (14.1%) with the second scale, *SSC-3A*; and 246 surveys (41.5%) were carried out with the *ACAS-12* scale, proposed in this article.

## Analysis method: Psychometric properties

As indicated, a psychometric analysis has been carried out of each version of the scale, the previous SSC-2A and SSC-3A scales, as well as the ACAS-12. These analyses begin with a univariate exploration of the performance of the items through their percentage distribution, with special attention to the missing values and central tendency (mean -M- and standard deviation-SD-) and position statistics (skewness -Sk- and kurtosis -K-).

The internal consistency of the items that make up the scale was assessed by calculating Cronbach's alpha coefficient, observing the homogeneity indices of the component items through the corrected item-total correlation ($r$) and the alpha value ($\alpha$) of the scale if the item was removed, in order to determine the suitability of deleting one of them ($r$ values $< 0.30$ or an increase of the alpha coefficient above 0.05 points will permit the removal of the item, as indicated by Hair et al. [37]). Likewise, the commonality of the items ($h^2$) was considered as components of a single factor of the scale, where $h^2$ would refer to the percentage of variance that the factor would explain for each item [38].

Given that the variables were ordinal, the analysis of the relationship between the items was performed using the polychoric correlation matrix using the Factor 9.2 program [39, 40]. The suitability of the correlation matrix was tested in order to be factored through the analysis of the matrix determinant and the Kaiser-Meyer-Olkin (KMO) test and Bartlett's sphericity test [38]. The eigenvalues were analysed, and the MAP test (Minimum Average Partial Test) and the parallel test were requested to determine the number of factors to retain [41].

The construct validity of *ACAS-12* was performed with covariance structure techniques [42, 43] using the EQS 6 program [44, 45] and testing a single-factor measurement model From the

polychoric correlation matrix, multivariate skewness and kurtosis were evaluated with Mardia's tests [46] and in case of deviation from normality (Mardia's coefficient > 5), the weighted least squares estimation method would be used with the robust methods proposed by Satorra and Bentler [43, 47]. For the evaluation of the model fit the Satorra-Bentler robust chi-squared test ($\chi^2_{SB}$) was used. Given that this test is generally affected by sample size and lacks normality [42, 43, 48], the following indicators were used as the decision criterion: for a parsimony index the normed chi-square test ($\chi^2_{SB}$/df), whose values must range between 1 and 3 [43, 48] was estimated; for absolute fit indices, *Goodness of Fit Index*, GFI $\geq$ 0.90; *Root Mean Square Residual*, RMSR $\leq$ 0.08; and *Root MSE of Approximation*, RMSEA $\leq$ 0.08, and for incremental fit indices *Normed Fit Index*, NFI $\geq$ 0.90; *Non-Normed Fit Index*, NNFI $\geq$ 0.90; *Comparative Fit Index*, CFI $\geq$ 0.90; and *Adjusted Goodness of Fit Index*, AGFI $\geq$ 0.90. For the interpretation of these indices, the criteria proposed by Schweizer [49] were used. Finally, the Wald and Lagrange tests were requested to eliminate variables or incorporate new relationships, respectively; and the composite reliability and its explained variance were calculated with Raykov's procedure [50].

## Back-translation of scale

*ACAS-12* was formerly designed in Spanish. To facilitate its use in English, an adaptation and translation process was carried out following the criteria of the *International Test Commission* [51, 52]. Six people with written and spoken proficiency in both languages (three English and three Spanish) were involved in the translation process using the Delphi method.

In the first step, three translators, two from English and one from Spanish, independently translated from Spanish to English. One member of the research team received the translations, compared the results and ranked the similarities and differences. This first group came together to reach a consensus by resolving differences and the first main document of the *ACAS-12* scale in English was created. In the second stage, this translated version of *ACAS-12* was sent independently to the other three translators to translate it back into Spanish (back-translation). As before, another team member worked out the similarities and differences and then reached a consensus with the translators to create a back-translated version of *ACAS-12* in Spanish.

Finally, all the components of the adaptation team reviewed the translation and back-translation process, verified the semantic and syntactic concordance between the source version in Spanish and the back-translated version, and agreed on the final version in English of *ACAS-12*.

To assess the level of the inter-rater agreement at each step of translation, the kappa coefficient was calculated [53, 54]. This index takes into account the random effect on the agreement.

The S4 File includes the final version of the *ACAS-12* scale in Spanish (source language) and English (target language), as well as its instructions for use.

The level of agreement obtained in both stages of translation was substantially high (kappa = 0.692 in stage 1 (ES>EN) and high 0.846 in stage 2 (EN>ES)) [53–55]. To resolve the discrepancies, the participants in the back-translation were asked to choose the most appropriate term of the existing ones. After this exercise, the level of agreement level was quickly achieved, reaching 0.916 of final convergence.

## Results: Study of psychometric evidence in the scale construction process

### Analysis of previous versions: Assessment scales with 2 and 3 attributes

The first version of the scale (*SSC-2A*) was used in stage 1 of the process. It was applied to evaluate seven places. 263 people participated in this first group of studies. The scale proposes

twelve qualities of the soundscape, and the participants must select two antonyms to evaluate each of the proposed dimensions. The data from this first set of studies appear in Table 2.

One notable result was the high non-response rate, which ranged from 7.6% (n = 20) in *Familiarity* to 29.3% (n = 77) in the assessment of the concepts of *Informational capacity* and *Entertainment*. Of the total number of participants, almost a third (31.5%) omitted the response to one of the dimensions, so the valid sample for the psychometric use of the tool would be reduced to a total of 180 participants.

From the analysis of valid responses (see Table 2, column '% Responses'), most of the dimensions assessed showed high percentages associated with the positive aspects of urban acoustic comfort. In six of the twelve dimensions, the response percentages were above 66%, which contribute to a significant ceiling effect that raised concerns about the discriminatory power of the formulation of the items.

Analysis of the polychoric correlation matrix revealed a determinant of 0.056 and a KMO coefficient of 0.69, which raises doubts on potential scale factorization. The reliability analysis, based on the KR-20 test, was 0.65, which is considered moderate [56]. These results led to decisions on the need to improve the scale and the measurement system of the urban acoustic comfort dimensions.

Stage 2 of the process to define the assessment tool used a second version of the scale, *SSC-3A*. It was applied to evaluate 2 places. 84 people participated in this second group of studies. The second version of the scale is a semantic differential scale with the same adjectives as in *SSC-2A*. The participants had to score each adjective with three response alternatives. The scale allows the option of neutral evaluation (neither an adjective nor another) concerning the adjectives proposed to evaluate the soundscape. The data from this second set of studies appear in Table 3.

In the application of *SSC-3A*, as it was based on a scale there were no missing values, but there was a high percentage of responses in the neutral attribute, i.e., the intermediate or undifferentiated assessment between the two antagonistic attributes that differentiate the soundscape. Qualities with mean values greater than 2 would indicate a greater positioning towards positive assessments of acoustic comfort, whereas values below 2 would indicate a greater anchoring towards negative assessments. There were no large deviations from univariate skewness (values lower than -1.25 or higher than 1.25), but there were cases of high kurtosis (*conversation* and *familiarity*; K > 1.25).

**Table 2. Response distribution (%) of the scale version with two measurement attributes (*SCC-2A*) (n = 263).**

| Concepts | | % Responses | | | % Non-responses |
|---|---|---|---|---|---|
| Pleasantness | Unpleasant | 30.5 | 69.5 | Pleasant | 15.2 |
| Relaxation | Stressful | 41.7 | 58.3 | Relaxing | 27.0 |
| Calmness | Noisy | 58.1 | 41.9 | Peaceful, calm | 12.9 |
| Clarity | Confused, chaotic | 40.6 | 59.4 | Clear, accurate | 19.4 |
| Entertainment | Boring | 27.4 | 72.6 | Fun | 29.3 |
| Naturalness | Artificial | 33.8 | 66.2 | Natural | 11.0 |
| Vibrancy | Monotonous | 49.8 | 50.2 | Lively | 22.1 |
| Conversation | Hinders | 31.6 | 68.4 | Facilitates | 21.7 |
| Informational capacity | No informative | 52.7 | 47.3 | Informative | 29.3 |
| Congruence | Inappropriate | 28.1 | 71.9 | Appropriate | 27.0 |
| Familiarity | Unknown | 11.9 | 88.1 | Familiar | 7.6 |
| Continuity | Intermittent | 28.9 | 71.1 | Uninterrupted | 10.6 |

**Table 3. Descriptive data of the scale version with three measurement attributes (*SSC-3A*) (polychoric matrices; n = 84).**

| Concepts | | Percentage distribution (1) | | | | Descriptive | | | |
|---|---|---|---|---|---|---|---|---|---|
| | | 1 | 2 | 3 | | M | SD | Sk | Ku |
| Pleasantness | Unpleasant | 16.7 | 58.3 | 25.0 | Pleasant | 2.07 | 0.64 | -0.32 | 0.08 |
| Relaxation | Stressful | 16.7 | 60.7 | 22.6 | Relaxing | 2.05 | 0.63 | -0.31 | 0.24 |
| Calmness | Noisy | 21.4 | 46.4 | 32.1 | Peaceful, calm | 2.09 | 0.72 | -0.33 | -0.64 |
| Clarity | Confused, chaotic | 9.5 | 47.6 | 42.9 | Clear, accurate | 2.39 | 0.66 | -0.69 | 0.22 |
| Entertainment | Boring | 19.0 | 69.0 | 11.9 | Fun | 1.92 | 0.55 | -0.36 | 0.98 |
| Naturalness | Artificial | 34.5 | 39.3 | 26.2 | Natural | 1.90 | 0.78 | 0.03 | -1.09 |
| Vibrancy | Monotonous | 36.9 | 48.8 | 14.3 | Lively | 1.76 | 0.68 | 0.16 | -0.60 |
| Conversation | Hinders | 7.1 | 75.0 | 17.9 | Facilitates | 2.08 | 0.49 | -0.39 | 2.37 |
| Informational capacity | No informative | 21.4 | 59.5 | 19.0 | Informative | 1.94 | 0.64 | 0.18 | -0.02 |
| Congruence | Inappropriate | 14.3 | 33.3 | 52.4 | Appropriate | 2.38 | 0.76 | -0.78 | -0.05 |
| Familiarity | Unfamiliar | 0.0 | 35.7 | 64.3 | Familiar | 2.60 | 0.48 | -1.48 | 3.46 |
| Continuity | Intermittent | 22.6 | 39.3 | 38.1 | Uninterrupted | 2.12 | 0.76 | -0.36 | -0.91 |

(1) 1 = choose the adjective on the left; 3 = choose the adjective on the right; 2 = neither the adjective on the right nor the adjective on the left;

M–Mean; SD–Standard Deviation; Sk–Skewness; Ku–Kurtosis

The internal consistency coefficient reached by the *SSC-3A* scale was 0.73, considered a moderate value [56]. However, the assessment of the suitability of the polychoric correlation matrix to be factored was not adequate with a determinant of 0.0059 and a KMO value of 0.68. Given the response distribution profile, excessively centred on neutral scores between the two adjectives, and the low consistency of the items with this response bias, it was decided to expand the measurement scale to a greater number of answer options, creating the *ACAS-12 scale* and test its psychometric suitability for the measurement of the acoustic comfort.

## Analysis of the final version of the scale: *ACAS-12*

Finally, the *ACAS-12* scale was used to evaluate 13 places. The scale is a 5-point semantic differential scale made up of twelve pairs of bipolar adjectives (stage 3). Table 4 presents the results obtained, with the items ordered according to their contribution to the scale as a whole. All the participants (n = 246) responded to all the items, except those of the Sestao square for which there was no data on Informational capacity. In the data of those sites missing values were replaced by the mean value of the range of the scale (3).

The analysis shows that there were no floor effects (option 1 percentages below 15%) but there were ceiling effects in all the items (option 5 percentages greater than 15%) [57], although not as strong as in the previous versions of the scale (with two -*SSC-2A*- and three attributes -*SSC-3A*-). Consequently, there was still a negative skewness (although not very sharp, between 0.0 and -0.96) of the highest percentage of participants' responses in positive assessments of urban acoustic comfort. However, while all means were above the value 3 (halfway from 1 to 5), only the item *familiarity* obtained an average value above 4.

In order to assess the feasibility of the factor analysis, the values of the polychoric correlation matrix of *ACAS-12* were analysed, finding a determinant near-zero (0.000608) and a KMO value of 0.92, considered very good. Likewise, Bartlett's sphericity test ($\chi^2$ = 1778.5, df = 66; p<0.001) indicated that the population correlation matrix was not an identity matrix. Therefore, it was considered feasible to factor in the correlation matrix.

Both the MAP test and the parallel test suggested the retention of a single factor [58, 59]. The analysis of the eigenvalues indicated the existence of the first factor with a value of 6.72,

**Table 4. Descriptive and internal consistency data of *ACAS-12* in its version with five measurement attributes (polychoric matrices; n = 246).**

| Concepts | | Percentage distribution (1) | | | | | | Descriptive | | | | Internal consistency | | |
|---|---|---|---|---|---|---|---|---|---|---|---|---|---|---|
| | | 1 | 2 | 3 | 4 | 5 | | M | SD | Sk | Ku | r | α | h² |
| Pleasantness | Unpleasant | 10.2 | 12.2 | 19.9 | 28.5 | 29.3 | Pleasant | 3.52 | 1.31 | -0.58 | -0.71 | 0.86 | 0.89 | 0.79 |
| Relaxation | Stressful | 6.1 | 7.7 | 24.0 | 31.7 | 30.5 | Relaxing | 3.71 | 1.15 | -0.76 | -0.04 | 0.83 | 0.89 | 0.78 |
| Calmness | Noisy | 13.8 | 13.0 | 20.3 | 26.0 | 26.8 | Peaceful, calm | 3.37 | 1.36 | -0.43 | -0.99 | 0.80 | 0.89 | 0.71 |
| Clarity | Confused, chaotic | 5.3 | 6.5 | 22.4 | 31.3 | 34.6 | Clear, accurate | 3.81 | 1.13 | -0.87 | 0.19 | 0.73 | 0.90 | 0.66 |
| Entertainment | Boring | 7.3 | 10.2 | 28.9 | 22.0 | 31.7 | Fun | 3.58 | 1.23 | -0.52 | -0.57 | 0.73 | 0.89 | 0.61 |
| Naturalness | Artificial | 14.2 | 8.9 | 15.9 | 21.5 | 39.4 | Natural | 3.61 | 1.43 | -0.68 | -0.86 | 0.69 | 0.90 | 0.63 |
| Vibrancy | Monotonous | 7.7 | 16.7 | 30.1 | 17.9 | 27.6 | Lively | 3.39 | 1.26 | -0.24 | -0.91 | 0.69 | 0.90 | 0.55 |
| Conversation | Hinders | 4.1 | 9.3 | 25.2 | 25.6 | 35.8 | Facilitates | 3.78 | 1.14 | -0.68 | -0.26 | 0.60 | 0.90 | 0.52 |
| Informational capacity | No informative | 8.9 | 4.5 | 56.5 | 9.3 | 20.7 | Informative | 3.26 | 1.12 | -0.12 | -0.04 | 0.52 | 0.90 | 0.38 |
| Congruence | Inappropriate | 2.8 | 11.8 | 18.3 | 45.5 | 21.5 | Appropriate | 3.69 | 1.04 | -0.79 | 0.25 | 0.50 | 0.91 | 0.34 |
| Familiarity | Unfamiliar | 2.8 | 2.0 | 26.0 | 27.6 | 41.6 | Familiar | 4.01 | 1.01 | -0.96 | 0.80 | 0.34 | 0.91 | 0.45 |
| Continuity | Intermittent | 5.7 | 5.3 | 21.5 | 36.6 | 30.9 | Uninterrupted | 3.80 | 1.10 | -0.95 | 0.49 | 0.34 | 0.91 | 0.46 |

(1) 1 = the adjective on the left describes it very well; 5 = the adjective on the right describes it very well; 3 = neither the adjective on the right describes it. Los values 2 and 4 correspond to fewer clear descriptions of the left and right adjectives respectively.

M–mean; SD–standard deviation; Sk–skewness; Ku–kurtosis; r–item-total correlation coefficient; α –total scale reliability coefficient if item is removed; h²: communality

much higher than that obtained by the second extracted factor, 1.08. This would explain 56% of the variance. Therefore, mathematically, everything seemed to indicate the existence of a single factor that would explain the set of relationships of the component items.

In the construction of the *ACAS-12* scale, the existence of a common factor was considered at a conceptual level (perception/evaluation of the acoustic environment, which is acoustic comfort). Therefore, to test the validity of the model a confirmatory factor analysis was conducted to corroborate whether the empirical data fit this construct. Although there were no major deviations from normality in the univariate analyses (Table 4), the lack of multivariate normality was confirmed. The Mardia's coefficients for skewness and kurtosis were 28.81 and 235.39, respectively.

Therefore, once the model was specified as a single factor that explains the 12 component items that fix the factor variance, the weighted least squares procedure was used as the robust estimation method. The results show a statistically significant Satorra-Bentler test ($\chi^2_{SB}$ = 155.52; df = 54; p<0.001) which would indicate that the data did not converge with the theoretical proposal imposed. Since this test is affected by the normality of the distribution and the sample size, other types of indicators were considered for the assessment of the model fit. Specifically, the normed chi-squared index ($\chi^2_{SB}$/df) gave a value of 2.88 (between 1 and 3 indicates suitability), and the absolute and incremental fit indices were above the minimum reference value of 0.90 (NFI = 0.97; NNFI = 0.97; CFI = 0.98; IFI = 0.98) and the residual-based indices showed values close to suitability (SRMR = 0.057; RMSEA = 0.042; 90% CI RMSEA = 0.002 to 0.066).

Table 5 presents the standardised factor coefficients (Lambda - λ) and the errors (Theta-Delta - δ) of the measurement model. All the factor loads were above the recommended value (>0.50), except for the *familiarity* (λ = 0.43) and *continuity* (λ = 0.44) items. However, Wald's analysis does not recommend dispensing with any of the items.

Given the lower contribution of these two items, it was decided to conduct a new confirmatory factor analysis specifying a unifactorial model made up of the 10 items with a higher contribution (that can be called *ACAS-10*), whose factor loads, and errors are shown in Table 5.

**Table 5. Confirmatory factor analysis of *ACAS-12* and *ACAS-10* (restricted) (n = 246).**

| Concepts | ACAS-12 | | ACAS-10 | |
|---|---|---|---|---|
| | λ | δ | λ | δ |
| Pleasantness | 0.915 | 0.403 | 0.904 | 0.428 |
| Relaxation | 0.902 | 0.433 | 0.904 | 0.427 |
| Calmness | 0.867 | 0.499 | 0.867 | 0.499 |
| Clarity | 0.808 | 0.589 | 0.815 | 0.580 |
| Entertainment | 0.800 | 0.600 | 0.798 | 0.602 |
| Naturalness | 0.777 | 0.629 | 0.745 | 0.607 |
| Vibrancy | 0.744 | 0.668 | 0.725 | 0.689 |
| Conversation | 0.678 | 0.736 | 0.694 | 0.720 |
| Informational capacity | 0.593 | 0.805 | 0.610 | 0.793 |
| Congruence | 0.541 | 0.841 | 0.526 | 0.850 |
| Familiarity | 0.430 | 0.903 | --- | --- |
| Continuity | 0.444 | 0.896 | --- | --- |

Notes: λ –Factor loading coefficients (Lambda); δ –Error coefficients (Theta-Delta)

The fit indices were also suitable for this model ($\chi^2_{SB}$ = 49.83; df = 35; p = 0.049; $\chi^2_{SB}$/df = 1.42; NFI = 0.99; NNFI = 0.99; CFI = 0.99; IFI = 0.99; SRMR = 0.034; RMSEA = 0.026; 90% CI RMSEA = 0.000 to 0.050).

Finally, from the standardised factor loads and applying Raykov's procedure the composite feasibility was estimated, which turned out to be 0.90 for both models, and the variance extracted was 44.21% for *ACAS-12* and 48.54% for *ACAS-10*. Therefore, both models could be suitable. However, it was preferred to maintain *ACAS-12* because it provided information on one of the acoustic comfort qualities that are highlighted in the literature (*Familiarity*).

The contrast of averages for the set of places where the *ACAS-12* scale (stage 3) was used has been significant both for the use of the *ACAS-mean*, according to the Brown-Forsythe robust test of equality of means (F = 26.45; df = 13;117.78; p<0.001), and for the *ACAS-fact* (F = 28.04; df = 13;126.35; p<0.001).

Considering the factor scores, positive means are observed for the calmest places (place 10 GLT_post and places 18 and 19 in Parque_Olarizu) and negative scores for the noisiest (places 11, 12 and 13 in Calle_Los_Herran; place 16 and place 17 in Plaza-_Constitución, and place 22 -south zones with benches- in Plaza_San_Pedro-Sestao). The global scores allow discrimination between places. In this sense, the Scheffe post-hoc test shows a statistically significant difference between the best place (place 18 -path- in Parque_Olarizu) and the worst place (place 11 near the bus-stop in Los_Herran) evaluated in Vitoria-Gasteiz (p <0.001) of +1.52 [95% CI: from +2.50 to +0.52].

The internal consistency analysis offered scale item-total correlation values above 0.30. In fact, 10 of the 12 items were above values of 0.50, and only two scales provided values close to 0.35 (*familiarity* and *continuity*, with r = 0.34). The Cronbach's alpha coefficient obtained in *ACAS-12* was 0.91, and the removal of none of the items would contribute to improving its reliability. Finally, Table 4 shows the commonality of the items when a single factor is retained, with a lower percentage of variance in the *information* ($h^2$ = 0.38) and *congruence* ($h^2$ = 0.34) items and all cases above the 0.45 value.

## Discussion

This article presents a proposal for a scale to measure acoustic comfort in urban settings: *ACAS-12*. It is a 5-point semantic differential scale structured around 12 pairs of opposite

adjectives. This proposal has been defined, considering the theoretical and practical publications consulted in the references, and the analysis of a database of environmental comfort assessments made in studies conducted on place, and carried out by the real users of the spaces. This database is considered diverse, since it covers a variety of types of open urban public places in which the main cultural functions are leisure and enjoyment (parks, squares, etc.), with diverse acoustic environments, participants (users), activities, times of the year and times of the day. Therefore, the database of assessments can represent a broad range of urban soundscapes.

To assess acoustic comfort, it is considered optimal to involve people who use the urban environments that are the object of the assessment. By conducting on-place surveys at sites of interest, the ecological validity of the studies is significantly improved, as is remarked in ISO 12913:2:2018 [19]. First, because it is applied in real urban environments; and second, because it collects the perceptual information of the real users of these urban environments. Furthermore, the analysis of acoustic and environmental comfort can be a tool that facilitates the participation of people in decision-making processes about acoustic environments in collaboration with "local experts" [60].

In this sense, the semantic differential, in addition to being the type of scale most used in soundscape studies, its presentation (visual aspect) facilitates its understanding and can be answered easily, clearly and quickly by the participants. Another notable aspect of the semantic differential is that it allows access to the content, whether of cognitive or connotative (emotional) components [1, 19], in our case acoustic comfort.

The application made in this research of psychometric analysis methods to assess the suitability of evaluation scales for soundscapes is considered novel. Moreover, the analysis carried out for the design of the scale shows the goodness of a scale with 5 response levels compared to one of 3 or fewer. The psychometric study has shown that scales with fewer response levels, 2 or 3, compared to the 5-point *ACAS-12*, produce a significant bias with strong skewness. Furthermore, 5-point scales provide a good balance between the theoretical quality of the scale and the friendliness of the assessment tool for the public. In the latest version of ISO 12913–2: 2018 [19], the Perceived Affective Quality and Appropriateness scales are also proposed in a 5-point ordinal category.

Another contribution of this proposal is the selected list of 12 pairs of adjectives, which are structured around the five dimensions mentioned in different previous research: *Pleasantness*, *Eventfulness* and *Familiarity*, as consolidated dimensions, adding *Informational capacity* and *Congruence*, considered by other authors. Regarding the description of the dimensions of *Pleasantness and Eventfulness*, the proposal follows the approach of other researchers by defining an orthogonal space between both dimensions. However, different pairs of adjectives are proposed to represent some of the dimensions concerning what is exposed in the soundscape standards (ISO, 2014, 2018, 2019). Besides, the scale includes two adjectives, grouped in the dimension *Pleasantness* that refer to the well-known positive perception of *nature* and the restorative capacity of the soundscape, called *relaxing* versus *stressful*. Similar adjectives, together with *pleasant*, were part of the first factor extracted in the soundscape analysis by Kawai et al. [61] called *preference* or *calmness*, according to Cain et al. [1].

The analysis of the *ACAS-12* scale has confirmed the high internal consistency of the selected items since it shows that the set of 12 items is comprehensive: a Cronbach's alpha coefficient of 0.91 and a composite reliability of 0.90 indicate high reliability and precision of the measurement scale. It is also shown that although there are two acoustic comfort qualities of *Familiarity* (*unknown-familiar*, *intermittent-uninterrupted*) and have a slightly different behaviour their elimination does not present greater goodness from the 12–item version.

It is novel that the results of the studies carried out with the *ACAS-12* scale indicate that the best factorial solution possible is that of a single factor that would explain around 50% of the *ACAS-12* variance (44% of extracted variance by Raykov's method or 56% by the MAP test). This means that the complete set of 12 items is necessary to form a general assessment factor for acoustic comfort. This result comes into the discussion with the conclusions reached in other recognized studies [2, 26, 28, 32]. It should be said that the result presented in this article is supported by the rigour of the method used for the analysis of the database generated in several projects. Moreover, the aggregated values of *ACAS-12* (averages and factorial scores) allow us to order the analysed places following the logic derived from the knowledge of the characteristics of the places and this is derived from the combination of several interrelated variables (not just one), such as nature content (blue and green), use, social interactions, acoustic environment. Authors such as Tarlao et al. [29] state that the generality of the two-dimensional model is being analysed. Anyhow there is a need for a future dialogue with other results. In this dialogue, it is worth paying attention to the analysis method and criteria to be used to evaluate the analysis tools of soundscapes, since the application of similar analyses between different research groups could give way to future research on the best factorial solution possible solution to understand the perception of sound environments [1, 26, 28, 32, 34].

Nevertheless, the result achieved in this research is that a single factor that explains the soundscape does not reduce the theoretical importance of soundscape dimensions to understand and categorise acoustic comfort or to define design strategies. On the contrary, it highlights the possible contribution of each pair of adjectives to evaluate and improve acoustic comfort.

On the other hand, the English version of the scale is created by translating the Spanish version used in the studies. The translation was elaborated with a scientifically recognised procedure. This seeks to ensure its transcultural validity and facilitate its consideration by other authors and contribute to the definition of a common scale. This is intended to contribute to the requirements for validation across languages and sites, which are also set out in part 3 of ISO 12913 on soundscape [20]. Therefore, while similar scales are defined in other languages, the naming of the adjective pairs should probably be understood as an open process.

Given all this, the *ACAS-12* scale can be considered a proposal for acoustic comfort assessment that aims to contribute to the scientific community with a valid, reliable tool for measuring the pleasantness of the acoustic environment in an urban setting, by taking a psychosocial assessment of the perception and not based on indirect objective indicators.

The goal of this kind of research is to contribute to the understanding of how an urban soundscape can create a pleasant environmental experience (acoustic and urban comfort) for people, contributing to the well-being and health of the population [62]. Therefore, this knowledge can be applied to adapt the design of new urban spaces, or the improvement of existing ones, to environmental conditions and the requirements of their users. Holistic urban planning and renewal can play a key role to improve well-being and health [63]. The ability of a place to enhance people's well-being is related to the concept of a restorative environment, i.e., environments that enhance or facilitate psychological restoration and thus contribute to human health and well-being [64].

In conclusion, the *ACAS-12* scale is a methodology proposed for conducting acoustic comfort studies in urban spaces. However, future work is still needed. Its validation is a long and complex process to be done. The authors will further analyse its performance in comparison with other scales (concurrent validity), and its suitability for different places, especially for places where a more negative environmental noise can produce distortion in positive perception or acoustic discomfort (discriminant validity). Likewise, the stability of the scale and its sensitivity to change will also be studied, as well as the possibility of integrating the temporal

dimension (days, weeks, months, seasons, etc.) into the measurement scale, which would be of great help when describing acoustic comfort though out a path (soundwalks). Furthermore, given that the studies considered in this research are urban, future studies will be necessary to analyse the validity of the *ACAS-12* tool for non-urban environments (peri-urban, rural, or natural) and the need to adapt it to the perception of these areas.

## Supporting information

**S1 File. *SSC-2A* scale database of stage 1 (n = 263).**
(XLS)

**S2 File. *SSC-3A* scale database of stage 2 (n = 84).**
(XLS)

**S3 File. *ACAS-12* scale database of stage 3 (n = 246).**
(XLS)

**S4 File. Acoustic Comfort Assessment Scale (*ACAS-12*) in Spanish (native language) and English.**
(DOCX)

## Acknowledgments

This study has been made possible thanks to the selfless participation of the public (users of the spaces analysed), who contributed by responding to the questionnaire with their perception of the urban spaces, and the people who participated in the back-translation process of the scale. We thank them for their invaluable contribution.

## Author Contributions

**Conceptualization:** Karmele Herranz-Pascual, Itziar Aspuru, Igone Garcia-Pérez, José Luis Eguiguren, Álvaro Santander.

**Data curation:** Karmele Herranz-Pascual, Ioseba Iraurgi.

**Formal analysis:** Karmele Herranz-Pascual, Ioseba Iraurgi.

**Funding acquisition:** Itziar Aspuru, Igone Garcia-Pérez, José Luis Eguiguren.

**Investigation:** Karmele Herranz-Pascual, Ioseba Iraurgi, Álvaro Santander.

**Methodology:** Karmele Herranz-Pascual, Ioseba Iraurgi, Itziar Aspuru, Igone Garcia-Pérez, José Luis Eguiguren.

**Project administration:** Karmele Herranz-Pascual, Itziar Aspuru, Igone Garcia-Pérez.

**Resources:** Karmele Herranz-Pascual, Itziar Aspuru, Igone Garcia-Pérez, José Luis Eguiguren.

**Supervision:** Karmele Herranz-Pascual, Ioseba Iraurgi, Itziar Aspuru, Igone Garcia-Pérez, José Luis Eguiguren, Álvaro Santander.

**Validation:** Ioseba Iraurgi.

**Visualization:** Karmele Herranz-Pascual, Igone Garcia-Pérez.

**Writing – original draft:** Karmele Herranz-Pascual, Ioseba Iraurgi, Itziar Aspuru, Igone Garcia-Pérez, José Luis Eguiguren.

**Writing – review & editing:** Karmele Herranz-Pascual, Ioseba Iraurgi, Itziar Aspuru, Igone Garcia-Pérez, José Luis Eguiguren, Álvaro Santander.

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
