## [Decision Letter · Decision Letter 0]

23 May 2022

PONE-D-21-30882Development of the Acoustic Comfort Assessment Scale (ACAS-12): Psychometric properties, validity evidence and back-translation between Spanish and EnglishPLOS ONE

Dear Dr. Herranz-Pascual,

Thank you for submitting your manuscript to PLOS ONE. After careful consideration, we feel that it has merit but does not fully meet PLOS ONE’s publication criteria as it currently stands. Therefore, we invite you to submit a revised version of the manuscript that addresses the points raised during the review process.

Please note that we have only been able to secure a single reviewer to assess your manuscript. We are issuing a decision on your manuscript at this point to prevent further delays in the evaluation of your manuscript. Please be aware that the editor who handles your revised manuscript might find it necessary to invite additional reviewers to assess this work once the revised manuscript is submitted. However, we will aim to proceed on the basis of this single review if possible.  The reviewer's assessment of your paper was broadly positive (see comments appended below). However, the reviewer has requested clarification on some aspects which would improve the manuscript if addressed - please ensure you respond to each point carefully in your response to reviewers, and revise your manuscript accordingly.

We look forward to receiving your revised manuscript.

Kind regards,

Joseph Donlan

Editorial Office

PLOS ONE

Journal Requirements:

2. Thank you for stating the following in the Acknowledgments Section of your manuscript: "The research presented in this manuscript has been performed with the financial support of the Basque Government, the Bilbao City Council, and the EU within the framework of the LIFE QUADMAP project (LIFE 10/ENV/IT/407) and the CITI-SENSE FP7 project (http://www.citi-sense.eu/). This study has been made possible thanks to the selfless participation of the general public (users of the spaces analysed), who contributed by responding to the questionnaire with their perception of the urban spaces, and the people who participated in the back-translation process of the scale. We thank them for their invaluable contribution."

Please remove any funding-related text from the manuscript and let us know how you would like to update your Funding Statement. Currently, your Funding Statement reads as follows: "The author(s) received no specific funding for this work"

5. Please ensure that you include a title page within your main document. We do appreciate that you have a title page document uploaded as a separate file, however, as per our author guidelines (http://journals.plos.org/plosone/s/submission-guidelines#loc-title-page) we do require this to be part of the manuscript file itself and not uploaded separately.

Reviewers' comments:

Reviewer's Responses to Questions

**Comments to the Author**

1. Is the manuscript technically sound, and do the data support the conclusions?

Reviewer #1: Yes

2. Has the statistical analysis been performed appropriately and rigorously? 

Reviewer #1: Yes

3. Have the authors made all data underlying the findings in their manuscript fully available?

Reviewer #1: No

4. Is the manuscript presented in an intelligible fashion and written in standard English?

Reviewer #1: Yes

5. Review Comments to the Author

Reviewer #1: Thank you for giving me the opportunity to review this interesting work. In my opinion this is a very well conducted research and the manuscript itself is very well written. Please find below some comments only that I consider minor but I believe can improve the quality of the paper:

1: In the methods section there are indices mentioned without a reference provided. I would strongly advise the authors to add those.

2: In the results section the authors re-state information that should be in method section, such as the cut-off criteria for the CFA fit indices.

3. In the results section, is best to first report on the factor structure (unidimensionality established) and then present reliability, as reliability assumes unidimensionality. It will make much more sense especially in this application.

4. Have the authors consider the prospect of adding measuremnt invariance explorations (say age gender etvc). If not why? I would recomend to add a MIMIC model for invariance in the study to make it even more sound methodologically.

I hope that these comments will be helpful.

6. PLOS authors have the option to publish the peer review history of their article (what does this mean?). If published, this will include your full peer review and any attached files.

Reviewer #1: No

---

## [Author Response · Author response to Decision Letter 0]

10 Oct 2022

We sincerely appreciate the reviewer's positive comments that contribute to the improvement of our manuscript. In the overall, we have taken into account all his/her suggestions. Below we briefly comment on the changes we have made following the suggestions we have considered more relevant and justify those we have not integrated into the manuscript.

In the first suggestion of the reviewer, he/she proposes referencing the indices mentioned in the methods section. The specific references have been included:

• new reference [38]: Costelo & Osborne (2005) in lines 302 and 306; and 

• new reference [43]: Osborne (2014) in lines 310, 314, 316 and 318.

Regarding the second suggestion, we believe that the results section describes and interprets the values of the indexes whose “cut off” points have already been explained or indicated in the methodology section. We have not found any mention of re-specification of the “cut offs”.

On the other hand, we have considered the reviewer's third suggestion. On that sense the results concerning reliability and internal consistence (lines 418 to 424) have been moved to the end (lines 482-488), once the unidimensionality of the scale has been addressed.

Finally, the reviewer suggests the possibility of testing the invariance of the results according to variables of interest. Among the variables collected in our study, we consider that there are no variables that allow this type of analysis to be carried out. One possibility would be to analyzing invariance according to the sex of the participants. But there is no scientific evidence that would allow us to assess a differential behavior between men and women in the soundscape. We believe that it would be more relevant to test the scale invariance when it is applied for soundscapes with different characteristics. For example, to compare urban and natural soundscapes. Based on the reviewer's suggestion we will consider this option in our future evaluations in which we will consider other soundscape contexts, urban versus rural, technological versus natural, etc.

We reiterate our thanks to the reviewer for his/her comments and suggestions, and hope that we have managed to respond to his/her considerations.

Kind regards

---

## [Decision Letter · Decision Letter 1]

15 Nov 2022

PONE-D-21-30882R1Development of the Acoustic Comfort Assessment Scale (ACAS-12): Psychometric properties, validity evidence and back-translation between Spanish and EnglishPLOS ONE

Dear Dr. Dr. Herranz-Pascual,

It is a fact that this work has been under review for a long time, prior to my involvement, and I can only assume how inconvenient this must have been for you. I reviewed your work, and also read the reviews by other reviewers and in my opinion this work is to be published subject to minor corrections. Please find below the comments of the second reviewer and please get back to me as soon as possible. I will make sure to treat the final manuscript as the highest priority.

Please include the following items when submitting your revised manuscript:A rebuttal letter that responds to each point raised by the academic editor and reviewer(s). You should upload this letter as a separate file labeled 'Response to Reviewers'.A marked-up copy of your manuscript that highlights changes made to the original version. You should upload this as a separate file labeled 'Revised Manuscript with Track Changes'.An unmarked version of your revised paper without tracked changes. You should upload this as a separate file labeled 'Manuscript'.If applicable, we recommend that you deposit your laboratory protocols in protocols.io to enhance the reproducibility of your results. Protocols.io assigns your protocol its own identifier (DOI) so that it can be cited independently in the future. For instructions see: https://journals.plos.org/plosone/s/submission-guidelines#loc-laboratory-protocols. Additionally, PLOS ONE offers an option for publishing peer-reviewed Lab Protocol articles, which describe protocols hosted on protocols.io. Read more information on sharing protocols at https://plos.org/protocols?utm_medium=editorial-email&utm_source=authorletters&utm_campaign=protocols.

We look forward to receiving your revised manuscript.

Kind regards,

Silia Vitoratou, Ph.D

Guest Editor

PLOS ONE

Journal Requirements:

Reviewers' 2 comments:

Reviewer's Responses to Questions

**Comments to the Author**

1. If the authors have adequately addressed your comments raised in a previous round of review and you feel that this manuscript is now acceptable for publication, you may indicate that here to bypass the “Comments to the Author” section, enter your conflict of interest statement in the “Confidential to Editor” section, and submit your "Accept" recommendation.

Reviewer #2: (No Response)

2. Is the manuscript technically sound, and do the data support the conclusions?

Reviewer #2: Partly

3. Has the statistical analysis been performed appropriately and rigorously? 

Reviewer #2: I Don't Know

4. Have the authors made all data underlying the findings in their manuscript fully available?

Reviewer #2: Yes

5. Is the manuscript presented in an intelligible fashion and written in standard English?

Reviewer #2: No

6. Review Comments to the Author

Reviewer #2: Thank you for the opportunity to review this interesting piece of research.

The authors have developed and validated a new instrument about Acoustic Comfort. My main points are the following:

• results of parallel test and MAP analysis are not clearly presented

• it is not clear from the current text if the EFA and CFA have been applied in the same dataset.

In addition, manuscript is poorly written, and it will be enormously beneficiated from an English typographical and grammatical review.

Thank you,

7. PLOS authors have the option to publish the peer review history of their article (what does this mean?). If published, this will include your full peer review and any attached files.

Reviewer #2: No

---

## [Author Response · Author response to Decision Letter 1]

12 Dec 2022

We sincerely appreciate the reviewer's positive comments that contribute to the improvement of our manuscript. 

The comments and suggestions of reviewer #2 have been followed by correcting and clarifying the text of the manuscript. Some additional clarifications are:

A. At the request of reviewer #2, some references have been added in the text to support the assessments of the results (lines 346, 374, 395, 411). They had not been added as previous version of manuscript because they correspond to conventional values that either appeared in the references cited methodological description or in classical manuals, such as the one added to the references [new 56].

B. In response to the comment about Kappa index valuation: The authors know that the magnitude of the kappa values can be influenced by two conditions: 1) the codes are equiprobable or their probabilities vary, and 2) whether the marginal probabilities for the two observers are similar or different. However, to take a point of reference, we have considered Cohen's standard criteria, which are reflected in many psychometrics manuals (e.g., Barbero-García, Vila-Abad, Holgado, 2015 [55]), introducing in the text of the article a nuance referring to the value 0.692 by considering it as “substantial”: "The level of agreement obtained in both stages of translation was substantial (kappa = 0.692 in stage 1 {ES>EN}) and high (0.846 in stage 2 {EN>ES}) [53-55]".

C. In response to the comment about the result of MAP and parallel test: For the factor analyses, the FACTOR program [39-40] is used to analyse the correlation matrix of the ACAS-12 items. Based on the theory developed by Velicer [41], this program provides the result of the MAP and parallel tests. This is a proposal of the number of factors to be retained. In our case, as indicated in the text (line 420), the result is a single factor. 

D. The reviewer asks us whether the same group of participants was used for the exploratory and confirmatory factor analysis, and the answer is yes. However, we would like to inform him of the reasons for this approach. Our a priori hypothesis is the existence of a single factor that is explained by the set of twelve facets of acoustic comfort as assessed by the 12 items of the ACAS, therefore, the main analysis is confirmatory factorisation, e.g., that the empirically observed data fit the proposed theoretical model. However, certain correlation matrix analysis procedures offered by the FACTOR program are used to provide further evidence of the timeliness of the factorisation (determinant close to zero, KMO > 0.80, and significant Bartlet's test of sphericity), of the assumption of one-dimensionality (MAP and Parallel tests, and criterion of the eigenvalue of the first factor greater than three times the eigenvalue of the second factor), and to obtain the communalities (h2), necessary for the calculation of the omega reliability index. Hence, the analyses of both factorisation processes come from the same group of participants.

We reiterate our thanks to the reviewer for his/her comments and suggestions, and hope that we have managed to respond to his/her considerations.

---

## [Editor Report · Decision Letter 2]

13 Dec 2022

PONE-D-21-30882R2Development of the Acoustic Comfort Assessment Scale (ACAS-12): Psychometric properties, validity evidence and back-translation between Spanish and EnglishPLOS ONE

Dear Dr. Herranz-Pascual,

Thank you for submitting your manuscript to PLOS ONE. After careful consideration, we feel that it has merit but does not fully meet PLOS ONE’s publication criteria as it currently stands. Therefore, we invite you to submit a revised version of the manuscript that addresses the points raised during the review process.

We look forward to receiving your revised manuscript.

Kind regards,

Silia Vitoratou, Ph.D

Guest Editor

PLOS ONE

Journal Requirements:

**Additional Editor Comments:**

Dear authors,

It is not methodologically acceptable to use EFA and CFA in the same data. This is a serious methodological error, even though I understand the rational of using different tests in your rebuttal letter. I suggest to retain the tests for suitability of using factor analysis (Bartlet, KMO etc) but delete MAP/Parallel analysis/EFA results. The CFA explorations you conduct are more than enough.

I hope this helps.

---

## [Author Response · Author response to Decision Letter 2]

12 Jan 2023

Rebuttal letter 2-bis

We sincerely appreciate the reviewer's positive comments that contribute to the improvement of our manuscript. 

The comments and suggestions of reviewer #2 have been followed by correcting and clarifying the text of the manuscript:

Comments and suggestions of reviewer #2 Authors' response

Abstract 

• Line 23: please change “assessment of human perception” to “human perception assessment” DONE

• Please be consistent: there are the terms five-point and 2-point and 3-point. Kindly use the full word or the number throughout. all the manuscript done and revised

• Line 35: kindly change: “The back-translation technique has been used for writing the English version.” to “the back-translation technique has been applied to create the English version” DONE

• L35-36: You mention that “The results highlight the good psychometric properties of the ACAS 12 scale” but no results have been presented in the abstract. I would suggest presenting some of your findings in the abstract to back-up this argument. Added: "…(Cronbach’s alpha 0.91 and composite reliability 0.90)”

• Line 36: “factorial solution possible”; please delete the word possible from this sentence DONE

• Line 37: please change the word “around” to “approximately” DONE

• L36: Current factor analysis techniques are mostly based on Goodness of fit statistics (i.e., RMSEA, CFI, TLI, etc.). Kindly provide these in the abstract to reinforce your argument that a one-factor solution is the most appropriate one. Added: "The absolute and incremental fit indices were above the minimum reference value of 0.90 and the residual-based indices showed values close to suitability (SRMR=0.057; RMSEA=0.042This).”

Text 

• Line 43: “This is why” please change to “this the reason for which” DONE

• Line 43: “have been mainly focused” please delete the word “been” DONE

• Lines 49-50: “The enhancement of comfort, in general, and acoustic comfort of urban spaces, in particular, entails considering them places for the enjoyment and relaxation of the people who use them” please rephrase, meaning not clear. Clarified: the sentence has been deleted

• Line 54: please change the word “proposed” to “suggested” DONE

• Line 67-68: “5) evaluation based on triangulation applying different methods” please rephrase meaning not clear. Clarified: “5) evaluation based on different methods of assessment (triangulation)”.

• Line 88: “Its applies” please correct DONE

• Line 95: “ACAS-12 is focus” please correct DONE

• Line 98: please change word “where” to “in which” DONE

• Line 128: please change to “obtained” DONE

• Line 134: sentence refers to past or in general? If in past, then kindly use past tense. DONE

• Line 152: please change the word “would” to “could” DONE

• Line 157: kindly rephrase; it is very difficult to understand the meaning of this sentence. Rewritten the first part of the first paragraph of methods section

• Line 346: please provide a reference about the kappa coefficient (why kappa=0.692 is characterised as high?) See RESPONSE A (NEW references [53, 54,55]) and RESPONSE B.

• Line 373: “casts doubt” kindly rephrase Rewritten

• Line 374: please provide relevant reference See RESPONSE A 

NEW reference [56].

• Table 3: please indicate what 1, 2, and 3 stand for DONE in the footer

• Line 395: please provide reference See RESPONSE A

NEW reference [56].

• Table 4: kindly provide in the footnotes what numbers 1-5 mean DONE in the footer

• Line 411: kindly provide a reference why percentages >15% constitute floor effect See RESPONSE A

NEW reference [57].

Major 

• Line 430: where can we find the results of the MAP and the parallel test? See RESPONSE C 

NEW references [58,59].

• Line 436: did you use the same dataset to perform the CFA? See RESPONSE D

Some additional clarifications (RESPONSES) are:

• RESPONSE A: At the request of reviewer #2, some references have been added in the text to support the assessments of the results (lines 346, 374, 395, 411). They had not been added as previous version of manuscript because they correspond to conventional values that either appeared in the references cited methodological description or in classical manuals, such as the one added to the references [new 56].

• RESPONSE B: In response to the comment about Kappa index valuation: The authors know that the magnitude of the kappa values can be influenced by two conditions: 1) the codes are equiprobable or their probabilities vary, and 2) whether the marginal probabilities for the two observers are similar or different. However, to take a point of reference, we have considered Cohen's standard criteria, which are reflected in many psychometrics manuals (e.g., Barbero-García, Vila-Abad, Holgado, 2015 [55]), introducing in the text of the article a nuance referring to the value 0.692 by considering it as “substantial”: "The level of agreement obtained in both stages of translation was substantial (kappa = 0.692 in stage 1 {ES>EN}) and high (0.846 in stage 2 {EN>ES}) [53-55]".

• RESPONSE C: In response to the comment about the result of MAP and parallel test: For the factor analyses, the FACTOR program [39-40] is used to analyse the correlation matrix of the ACAS-12 items. Based on the theory developed by Velicer [41], this program provides the result of the MAP and parallel tests. This is a proposal of the number of factors to be retained. In our case, as indicated in the text (line 420), the result is a single factor. 

• RESPONSE D: The reviewer asks us whether the same group of participants was used for the exploratory and confirmatory factor analysis, and the answer is yes. However, we would like to inform him of the reasons for this approach. Our a priori hypothesis is the existence of a single factor that is explained by the set of twelve facets of acoustic comfort as assessed by the 12 items of the ACAS, therefore, the main analysis is confirmatory factorisation, e.g., that the empirically observed data fit the proposed theoretical model. However, certain correlation matrix analysis procedures offered by the FACTOR program are used to provide further evidence of the timeliness of the factorisation (determinant close to zero, KMO > 0.80, and significant Bartlet's test of sphericity), of the assumption of one-dimensionality (MAP and Parallel tests, and criterion of the eigenvalue of the first factor greater than three times the eigenvalue of the second factor), and to obtain the communalities (h2), necessary for the calculation of the omega reliability index. Hence, the analyses of both factorisation processes come from the same group of participants.

In addition, a revision of the English language has been carried out.

We reiterate our thanks to the reviewer for his/her comments and suggestions, and hope that we have managed to respond to his/her considerations.

---

## [Editor Report · Decision Letter 3]

26 Jan 2023

Development of the Acoustic Comfort Assessment Scale (ACAS-12): Psychometric properties, validity evidence and back-translation between Spanish and English

PONE-D-21-30882R3

Dear Dr. Herranz-Pascual,

We’re pleased to inform you that your manuscript has been judged scientifically suitable for publication and will be formally accepted for publication once it meets all outstanding technical requirements.

Kind regards,

Silia Vitoratou, Ph.D

Guest Editor

PLOS ONE

Additional Editor Comments (optional):

In my opinion the authors have sufficiently responded to the comments. I recommend publication of the revised manuscript.

---

## [Editor Report · Acceptance letter]

27 Jan 2023

PONE-D-21-30882R3 

Development of the Acoustic Comfort Assessment Scale (ACAS-12): Psychometric properties, validity evidence and back-translation between Spanish and English 

Dear Dr. Herranz-Pascual:

I'm pleased to inform you that your manuscript has been deemed suitable for publication in PLOS ONE. Congratulations! Your manuscript is now with our production department. 

Kind regards, 

on behalf of

Dr. Silia Vitoratou 

Guest Editor

PLOS ONE